# The Minor Matrix Protein VP24 from Ebola Virus Lacks Direct Lipid-Binding Properties

**DOI:** 10.3390/v12080869

**Published:** 2020-08-08

**Authors:** Yuan Su, Robert V. Stahelin

**Affiliations:** Department of Medicinal Chemistry and Molecular Pharmacology and the Purdue Institute for Inflammation, Immunology and Infectious Disease, Purdue University, West Lafayette, IN 47907, USA; su205@purdue.edu

**Keywords:** Ebola virus, filovirus, lipid binding, matrix protein, VP24

## Abstract

Viral protein 24 (VP24) from Ebola virus (EBOV) was first recognized as a minor matrix protein that associates with cellular membranes. However, more recent studies shed light on its roles in inhibiting viral genome transcription and replication, facilitating nucleocapsid assembly and transport, and interfering with immune responses in host cells through downregulation of interferon (IFN)-activated genes. Thus, whether VP24 is a peripheral protein with lipid-binding ability for matrix layer recruitment has not been explored. Here, we examined the lipid-binding ability of VP24 with a number of lipid-binding assays. The results indicated that VP24 lacked the ability to associate with lipids tested regardless of VP24 posttranslational modifications. We further demonstrate that the presence of the EBOV major matrix protein VP40 did not promote VP24 membrane association in vitro or in cells. Further, no protein–protein interactions between VP24 and VP40 were detected by co-immunoprecipitation. Confocal imaging and cellular membrane fractionation analyses in human cells suggested VP24 did not specifically localize at the plasma membrane inner leaflet. Overall, we provide evidence that EBOV VP24 is not a lipid-binding protein and its presence in the viral matrix layer is likely not dependent on direct lipid interactions.

## 1. Introduction

Ebola virus (EBOV) is a lipid-enveloped filovirus with a non-segmented negative single-strand RNA genome that belongs to the order *Mononegavirales*. The infection of EBOV in humans can cause a highly fatal Ebola virus disease (EVD), which is characterized by acute symptoms including hemorrhagic fever, severe fatigue, emesis and multiorgan dysfunction. The high mortality of EVD was exemplified with over 11,000 deaths during the outbreak in West Africa between 2013 and 2016 [1,2] and a recent outbreak in the Democratic Republic of the Congo that claimed more than 2200 lives [3]. While an EBOV vaccine gained FDA approval late in 2019, there is still much concern regarding EBOV outbreaks as there are a lack of approved treatments for those that become infected.

The genus *Ebolavirus* consists of six species: *Taï Forest ebolavirus* (Taï Forest virus, TAFV), *Reston ebolavirus* (Reston virus, RESTV), *Sudan ebolavirus* (Sudan virus, SUDV), *Bundibugyo ebolavirus* (Bundibugyo virus, BDBV), *Bombali ebolavirus* (putatively Bombali virus, BOMV) and *Zaire ebolavirus* (Ebola virus, EBOV) [4,5]. Among these, EBOV is studied most intensively as it is the main circulating pathogenic ebolavirus found in infected patients in the majority of outbreaks [6].

The approximately 19 kb RNA genome of EBOV encodes at least seven proteins: the nucleoprotein (NP), the polymerase cofactor viral protein 35 (VP35), the matrix protein VP40, the transmembrane glycoprotein (GP), the transcriptional activator VP30, VP24 and the RNA-dependent RNA polymerase L [7]. The core of the Ebola virions nucleocapsid is composed of NP, VP35, VP30, VP24 and L, and encapsulates the RNA genome, while the transmembrane GP mediates virus entry and VP40 facilitates virus budding and egress at the host plasma membrane [8,9,10].

EBOV VP24 was first characterized as a minor membrane-associated protein when it was observed excluded from the nucleocapsid. The authors treated sucrose-gradient-purified Sudan viruses with 1% NP-40 lysis buffer containing various NaCl concentrations (e.g., 0.05, 0.15 and 1 M). The supernatant and pellet fractions were collected and examined by SDS-PAGE (sodium dodecyl sulfate-polyacrylamide gel electrophoresis) after centrifuging with a 30% sucrose cushion. The results showed that VP24 was not present in the pellet under medium (0.15 M) and high (1 M) salt concentrations whereas the nucleocapsid marker NP was in the pellet only, which lead to the assumption that VP24 (7.5% of virion protein) may behave as a secondary membrane associated protein like the matrix protein VP40 (37.7% of virion protein). However, VP24 was detected in both supernatant and pellet under low (0.05 M) salt conditions, the origins of which were not addressed in the previous report [11].

EBOV VP24 was also suggested to be a minor matrix protein when it was found that its cellular localization was distributed predominantly at perinuclear regions with a minor population on the plasma membrane in COS-1 cells [12]. Furthermore, the majority of VP24 protein expressed in human HEK 293T cells was found to be in the detergent phase through Trion X-114 mediated phase separation [12], where membrane associated proteins should accumulate. Lastly, VP24 was detected in the fraction collected via virus-like particle (VLP) purification method and sensitive to trypsin only in the presence of Triton X-100, indicating VP24 is incorporated into VLPs (or some lipid-encapsulated vesicle) [12]. Hence, EBOV VP24 may be a lipid-binding protein; however, the direct contribution of VP24 to VLP budding is not well understood [13].

While the lipid binding and role of VP24 in the viral matrix layer is unknown, a number of studies have shed light on the roles of VP24 in the regulation of viral transcription and replication, nucleocapsid assembly and transport, and interferon (IFN)-mediated immune signaling [14,15,16,17,18,19,20,21,22,23,24,25]. By utilizing the EBOV minigenome system, VP24 was found to inhibit viral genome transcription and replication [14,15]. VP24 is also vital for the nucleocapsid formation, condensation/maturation, transport, and thus yielding a functional virion by interacting with NP and VP35 [15,16,17,18,19,20,25]. Besides VP35, VP24 can suppress the innate immune response by preventing the nuclear accumulation of IFN-induced tyrosine-phosphorylated transcription factor STAT1 via binding host karyopherin *α* proteins which are nuclear localization signal receptors of activated STAT1 [21,22,23,24].

In contrast, the typical matrix proteins in filovirus such as EBOV VP40 (eVP40) and Marburg virus (MARV) VP40 (mVP40) are membrane associated peripheral proteins that bind to membrane lipids directly with high affinity. For instance, the association of eVP40 with phosphatidylserine (PS) and phosphatidylinositol-4,5-biphosphate (PI(4,5)P_2_) in the inner leaflet of the plasma membrane facilitates extensive eVP40 oligomerization and virus particle budding [26,27,28,29], while mVP40 tends to bind anionic membrane lipids nonspecifically through electrostatic interactions [30,31,32,33]. The expression of either eVP40 or mVP40 in mammalian cells is also sufficient to induce VLP generation [13,34,35,36].

To date, whether VP24 associates with membrane lipids is still unknown. To gain more direct evidence, we applied several lipid-binding assays to assess the lipid-binding ability of VP24. We also investigated the effect of potential posttranslational modification and the presence of eVP40 on VP24–lipid interactions. Furthermore, the protein–protein interaction between VP24 and VP40 was investigated by co-immunoprecipitation (Co-IP) and confocal fluorescence imaging analysis. The results we present here demonstrate that Ebola VP24 is not a direct lipid-binding protein and may not serve as a minor matrix protein with respect to lipid–protein interactions.

## 2. Materials and Methods

### 2.1. Plasmids

The VP24 plasmid for protein expression in *E. coli* pET46-His-VP24 with hexahistidine tag at the amino terminus and in mammalian cells pCAGGS-VP24-mCherry, were synthesized by Epoch Life Science, Sugar Land, TX, USA. Protein expression constructs pET46-His-eVP40, pET46-His-mVP40, pcDNA3.1-EGFP-eVP40, and pC1-LactC2-EGFP were described previously [28,30]. The plasmid pcDNA3-HA-VP24 is from BEI Resources (BEI Resources NR-49203) and the pEGFP-N1-EGFP-eVP24 plasmid was a kind gift from Dr. Julian A. Hiscox (University of Liverpool, UK).

### 2.2. Lipid Stocks

All lipids were purchased from Avanti Polar Lipids, Inc. (Alabaster, AL, USA), including 1,2-dioleoyl-sn-glycero-3-phosphocholine (DOPC), 1-palmitoyl-2-oleoyl-sn-glycero-3-phosphoethanolamine (POPE), 1-palmitoyl-2-oleoyl-sn-glycero-3-phospho-L-serine (POPS), 1-palmitoyl-2-oleoyl-sn-glycero-3-phosphate (POPA), 1,2-dioleoyl-sn-glycero-3-phospho-(1′-myo-inositol-3′-phosphate) (PI(3)P), 1,2-dioleoyl-sn-glycero-3-phospho-(1′-myo-inositol-4′-phosphate) (PI(4)P), 1,2-dioleoyl-sn-glycero-3-phospho-(1′-myo-inositol-5′-phosphate) (PI(5)P), 1,2-dioleoyl-sn-glycero-3-phospho-(1′-myo-inositol-3′,4′-bisphosphate) (PI(3,4)P_2_), 1,2-dioleoyl-sn-glycero-3-phospho-(1′-myo-inositol-3′,5′-bisphosphate) (PI(3,5)P_2_), 1,2-dioleoyl-sn-glycero-3-phospho-(1′-myo-inositol-4′,5′-bisphosphate) (PI(4,5)P_2_) and 1-stearoyl-2-arachidonoyl-sn-glycero-3-phospho-(1′-myo-inositol-3′,4′,5′-trisphosphate) (PI(3,4,5)P_3_). Lipids were generally stored in chloroform at −20 °C, with the exception of phosphoinositides, which were dissolved and stored in chloroform:methanol:water (20:9:1, *v*/*v*/*v*).

### 2.3. Antibodies

The antibodies used in this study include monoclonal anti-polyhistidine−alkaline phosphatase antibody (Milliposre Sigma #A5588, St. Louis, MO, USA), anti-EGFP monoclonal antibody (ThermoFisher Scientific #MA1-952, Grand Island, NY, USA), anti-HA tag antibody (Abcam #ab18181, Cambridge, MA, USA), anti-EBOV VP24 polycolonal antibody (IBT Bioserveices #0301-047, Rockville, MD, USA), anti-EBOV VP30 polycolonal antibody (IBT Bioserveices #0301-048, Rockville, MD, USA), anti-EBOV VP40 polycolonal antibody (IBT Bioserveices #0301-010, Rockville, MD, USA), anti-GAPDH monoclonal antibody (6C5) (ThermoFisher Scientific #AM4300, Grand Island, NY, USA), anti-alpha 1 sodium potassium ATPase antibody (Abcam #ab7671, Cambridge, MA, USA), anti-LAMP2 antibody (Abcam #ab13524, Cambridge, MA, USA), and anti-histone H3 antibody (Abcam #ab24834, Cambridge, MA, USA).

### 2.4. Cell Culture and Transfection

Human embryonic kidney 293 (HEK293) cells (from American Type Culture Collection, Manassas, VA, USA) were grown and maintained in Dulbecco′s Modified Eagle’s Medium (DMEM) containing l-glutamine, 4.5 g/L glucose and sodium pyruvate (Fisher scientific #MT10013CV, Waltham, MA, USA) containing 10% fetal bovine serum (FBS) and 1% Penicillin-Streptomycin (Fisher scientific #15-140-122, Waltham, MA, USA) at 37 °C in a humidified incubator supplied with 5% CO_2_. The transfection of HEK 293 cells was mediated with lipofectamine LTX reagent (Thermo Fisher Scientific #15338100, Waltham, MA, USA) when the cell confluency reached ~70% according to the manufacturer′s instructions.

### 2.5. Protein Expression and Purification

The protein expression constructs pET46-His-VP24, pET46-His-eVP40 and pET46-His-mVP40 were transformed into *E. coli* strain BL21 (DE3) pLysS. *E. coli* were grown at 37 °C in Luria Broth and 0.5 mM of isopropyl β-D-1-thiogalactopyranoside (IPTG) was added into the *E. coli* culture when the O.D. at 600 nm was ~0.6 to induce the protein expression. The cells were collected after 12–14 h of growth at 16 °C for VP24, and at room temperature (RT) for eVP40 and mVP40. The protein purification was performed through affinity binding of the hexahistidine tag with nickel-nitrilotriacetic acid (NTA) beads (Qiagen #30230, Valencia, CA, USA) following the manufacturer′s procedures.

### 2.6. Protein–Lipid Overlay Assay

PIP strips (Echelon biosciences #P-6001, Salt Lake City, UT, USA) and membrane lipid strips (Echelon biosciences #P-6002, Salt Lake City, UT, USA) are hydrophobic membranes containing 15 different lipids (≥C16) at 100 pmol per spot along with a solvent control spot. The detailed information of each lipid species can be found on the product′s website (also see Figure 1). The lipid strips were first blocked with 3% fatty acid-free bovine serum albumin (BSA) (Fisher scientific #BP9704100, Waltham, MA, USA) in Tris-buffered saline, 0.1% Tween 20 (TBST) (20 mM Tris-HCl, pH 8.0, containing 150 mM NaCl) for 1 h at RT followed by three washes with adequate TBST buffer for 5–10 min per time. Then, 1 μg/mL of purified recombinant protein in TBST was incubated with the lipid strip overnight at 4 °C with gentle agitation. When proteins extracted from HEK293 cells were used, cells were lysed in NP-40 lysis buffer (50 mM Tris-HCl, pH 7.4, containing 150 mM NaCl, 0.1% NP-40, 1× Halt protease inhibitors) and 80 μg/ml cell lysate in TBST was incubated with lipid strips. Next, the protein solution was discarded and the lipid strip was washed three times with TBST. Afterwards, the lipid strip was incubated with monoclonal anti-polyhistidine-alkaline phosphatase antibodies (1:5000–1:10,000 dilution) in TBST for 1 h at RT. Three washes with TBST were carried out before adding alkaline phosphatase substrates (Bio-rad #1706432, Hercules, CA, USA) to detect lipid bound protein.

### 2.7. Protein–Liposome Sendimentation Assay

To perform lipid-binding assays using a liposome sedimentation assay, liposomes were prepared using the following lipids compositions: PC:PE (70:30 mol ratio), PC:PE:PS or PA (70:20:10), and PC:PE:PIP (70:25:5). The lipid mixture was dried under a stream of nitrogen gas to completely dry the solvent (~30 min). The dried lipid films were then rehydrated in the extrusion buffer (25 mM HEPES, pH 7.5, containing 250 mM raffinose pentahydrate, 1 mM DTT) and incubated at 37 °C for at least 30 min with occasional vortexing during the incubation. The resuspended lipid solution was extruded through a 0.2-μm polycarbonate membrane 17 times by a lipid extruder to generate homogenous liposomes. The liposomes were collected after centrifuging at 50,000× *g* for 20 min at RT. The pelleted liposome was resuspended in the binding buffer (25 mM HEPES, pH 7.5, containing 150 mM NaCl, 0.5 mM EDTA, 1 mM DTT) and incubated with 0.5 μg of purified protein at RT for 45 min. The protein–liposome mixture was then centrifuged at 16,000× *g* for 30 min at RT to separate liposomes bound and free protein. Following centrifugation, the supernatant fraction was collected and the liposome pellet was resuspended in a 300–500-mL binding buffer and transferred into a new centrifuge tube subjected to another 30 min of centrifugation at 16,000× *g* at RT. Finally, the liposome pellet was collected and target protein was detected using Western blot in both supernatant and pellet fractions to assess the protein–liposome association.

### 2.8. Co-Immunoprecipitation

HEK293 cells were co-transfected with the HA-VP24 and VP40 plasmids and lysed with NP-40 lysis buffer (50 mM Tris-HCl, pH 7.4, containing 150 mM NaCl, 0.1% NP-40, 1× Halt protease inhibitors (Fisher scientific # PI78430, Waltham, MA, USA)) at 24–48 h post-transfection (hpt). 1 mg of cell lysate was incubated with 50 μL of 50% protein A bead slurry at 4 °C for 30 min and then centrifuged at 1000 rpm for 3 min at 4 °C to pre-clean the lysate. The pre-cleaned supernatant collected and incubated with 5 μg of antibodies at 4 °C overnight with gentle rocking. Protein A sepharose beads were added into the cell lysate and incubated for 1–4 h with agitation. The mixture was centrifuged at 500–1000 rpm for 30 s at 4 °C and then the beads were washed 3–4 times with 1 mL lysis buffer or PBS containing 0.2% Tween 20. The Laemmli buffer was added into the beads and the samples were subject to Western blot for analysis.

### 2.9. Subcellular Fractionation

HEK293 cells were collected 24 hpt and lysed with a plasma membrane protein extraction kit (Fisher scientific # NC1053480, Waltham, MA, USA). The separation of nuclei, cytosol, organelles and plasma membrane was performed according to the manufacture′s protocol.

### 2.10. Confocal Imaging

HEK293 cells were transfected with indicated plasmids and imaged at 24 hpt under a 100× oil objective on Zeiss LSM 880 upright confocal. Wheat germ agglutinin-Alexa Fluor 647 (WGA647) (Fisher scientific #W32466, Waltham, MA, USA) staining was performed based on the manufacture′s protocol. Image J software was used to analyze the fluorescent signal of different channels. Basically, a straight line was drawn across the plasma membrane and the fluorescence signal along the line was measured by the plot profile function.

## 3. Results

### 3.1. EBOV VP24 Does Not Associate with Lipids

EBOV VP24 has been suggested to be a minor matrix protein and peripheral protein as the virus or cellular fraction containing the VP24 protein was sensitive to the presence of detergent [11,12]. However, to the best of our knowledge, no further direct lipid-binding studies have been examined to determine how VP24 is recruited to the virus matrix layer and plasma membrane inner leaflet. In order to determine if VP24 is a bona fide lipid-binding peripheral protein, we performed several classical lipid-binding assays. First, two types of lipid strips were used: the PIP strip containing 15 different lipids at 100 pmol per spot, which features all seven phosphoinositides (PIPs) along with a solvent control spot. Second, the membrane lipid strip possessing 15 major membrane lipids in a similar setting to the PIP strip (Figure 1; each lipid species information—see Materials and Methods section), which covers the majority of bioactive lipids.

The lipid strips were blocked with 3% fatty acid-free bovine serum albumin (BSA) and then incubated overnight with 1 μg/mL of purified recombinant 6× His-tagged VP24 (His-VP24) protein or 6× His-tagged eVP40 or mVP40 protein as positive controls at 4 °C. The lipids strips were thoroughly washed afterwards, and the lipid-bound protein was detected by monoclonal anti-polyhistidine-alkaline phosphatase antibodies. The results exhibited lack of detectable binding by VP24 to lipids on either the PIP strip or the membrane lipid strip. In contrast, eVP40 and mVP40 bound significantly to a number of anionic lipids, as previously reported, including PS [27,29,37] and phosphoinositides [28,30,32,33,38]. Lastly, no background of lipid binding was observed using purified protein from *E. coli* embracing an empty expression vector (Figure 1).

### 3.2. VP24 Protein Produced in Human Cells Also Does Not Associate with Lipids

Next, we hypothesized that VP24 lipid-binding may be more dependent on mammalian cell expression due to potential post-translational modifications (PTMs) that may alter VP24–lipid interactions. For instance, VP24 is known to undergo SUMOylation that covalently attaches the small ubiquitin-like modifier (SUMO) to the target protein [39]. Thus, HEK293 cells were transfected with plasmids of HA-tagged VP24 (HA-VP24) or enhanced green fluorescent protein (EGFP)-tagged lactadherin C2 domain (LactC2) (LactC2-EGFP) which is a well-established PS binding protein [40]. First, the protein expression of HA-VP24 and LactC2-EGFP from HEK293 cells was examined (Figure 2A) demonstrating robust soluble protein expression. Next, the membrane lipid strips described above were incubated overnight at 4 °C with 80 μg/mL of cell lysate of HEK 293 cells, HEK 293 cells expressing HA-VP24 or HEK 293 cells expressing LactC2-EGFP. The primary anti-HA and anti-GFP antibodies along with the correlated secondary antibodies were applied to detect lipid-bound HA-VP24 and LactC2-EGFP, respectively. Similar to the mock control, no lipid-bound HA-VP24 was observed, while LactC2-EGFP bound to PS specifically as expected (Figure 2B), indicating that mammalian expression of VP24 had little impact on its lipid-binding ability.

### 3.3. The Major Matix Protein VP40 Does Not Facilitate the Membrane Association of VP24

Since VP24 does not seem to bind lipids independently, we asked whether the presence of VP40, the major matrix protein from EBOV, enhanced membrane affinity of VP24 through protein–protein interactions. Hence, we performed protein–liposome sedimentation assays using purified recombinant His-VP24 protein alone or a mixture of His-VP24 and His-VP40 incubated with liposomes containing different anionic lipids. All the liposomes had a composite of 70% (mol%) DOPC as the main structural component and varying percentages of POPE and anionic lipids of interest, as indicated in Figure 3. Since a high-density raffinose solution is constrained in the liposomes, the liposomes are pelleted following centrifugation. Therefore, the liposome-bound protein can be detected in the liposome pellet by Western blot. Similar to the protein–lipid overlay result, VP24 was absent in all the lipid-binding liposome pellets (Figure 3A). This was the case even in the presence of VP40 (Figure 3B). By contrast, VP40 was detected in the liposomes containing PIPs as previously reported [28,38].

To further investigate the relationship between VP24 and VP40, Co-IP assays were carried out to test whether VP24 interacts with VP40 directly or indirectly in cells. The HEK293 cells were co-transfected with equal amounts of HA-VP24 plasmid and non-tagged VP40 plasmid. Anti-VP24 and anti-VP40 antibodies were added to the cell lysate followed by protein A sepharose beads to pull down target proteins. An Anti-VP30 antibody was used as a control. The results revealed that the anti-VP24 and anti-VP40 antibodies successfully pulled down VP24 and VP40 protein, respectively, while anti-VP30 did not. However, VP24 was not detected in the VP40 precipitate and vice versa, suggesting no significant protein–protein interactions occurred between VP24 and VP40 (Figure 4).

### 3.4. Assessment of VP24 Cellular Localization

To assess the possibility of intracellular membrane association of VP24, a subcellular fractionation assay was utilized. Four cellular fractions were collected using the commercial isolation kit: nuclei, cytosol, organelles and plasma membrane from HEK293 cells expressing HA-VP24. Anti-GAPDH, anti-sodium potassium ATPase, anti-LAMP2 and anti-Histone H3 were used to check the purity and specificity of the cytosol, plasma membrane, organelle membrane and nuclei fractions, respectively. HA-VP24 was detected in the cytosol, organelle fraction, nuclei fraction, but barely in the plasma membrane fraction (Figure 5). However, there may be some slight contamination among the fractions, but the plasma membrane fraction was strikingly clean per the Western blot controls. Although a weak signal of cytosol marker GAPDH showed up in the plasma membrane fraction, it is established that GAPDH can partially localize to the plasma membrane [41].

To further validate the cellular localization of VP24, live cell fluorescence imaging analysis of VP24-mCherry or EGFP-VP24 was conducted. When transiently expressed in HEK 293 cells, two forms of VP24-mCherry were observed at 24 hpt: a cytoplasmic distribution while some localized at punctate structures (Figure 6). When VP24 was co-expressed with EGFP-VP40, which is plasma membrane localized, the plot profile of fluorescent intensity showed that VP24-mCherry did not colocalize with EGFP-VP40. Further, when labeling cells expressing EGFP-VP24 with plasma membrane marker wheat germ agglutinin, Alexa Fluor 647 (WGA-647) conjugate, EGFP-VP24 did not colocalize with WGA-647 specifically (Figure 6).

## 4. Discussion

Despite significant studies elucidating the role of VP24 as a viral transcription and replication modulator involved in nucleocapsid assembly and transport, as well as the role of VP24 as an interferon antagonist [8,10], recent studies refer to VP24 as the minor matrix protein [39]. In this study, we aimed to address the lipid-binding properties of VP24 and its role as a minor matrix protein to clarify its role in viral budding and its true role as a peripheral protein. First, we present direct evidence that purified recombinant VP24 protein does not associate to detectable levels with many bioactive lipids involved in viral assembly and budding as well as a host of typical interactions between peripheral proteins and membranes.

However, we could not rule out the possibility that VP24–lipid interactions may occur more readily under expression conditions of human cells or where VP24 undergoes PTMs. For example, the lipidation of proteins can facilitate the membrane recruitment of some proteins [42]. Thus, we repeated the lipid-binding assays with VP24 protein expressed in human cells where PTMs of VP24 should occur. The results also showed VP24 lacked detectable binding to membrane lipids (Figure 2), indicating PTMs do not enhance VP24 membrane association.

Thus, the lack of VP24 association with lipids made us wonder if it is possible that VP24 associates more readily with membranes due to interactions with the major matrix protein VP40. Protein–liposome sedimentation assays were used to assess VP24–lipid association in the presence of VP40 as liposomes provide a 3D membrane structure and curvature more representative of biological membranes than that of the lipid strips. Under these conditions, VP24 did not bind to any liposomes whereas VP40 bound liposomes containing PIPs (Figure 3). Notably, for the positive VP40 control PS association in the liposome pellet was not detectable, which is likely due to the relatively low amount of liposome (200 nmol) and low percentage (10%) of PS incorporated in the liposome in the assay. Notably, the efficiency of PS-VP40 binding critically depends on the percentage of PS in the liposome [37]. In a similar fashion to in vitro results, VP40 expression did not alter VP24 cellular localization. Thus, VP24 lipid interactions do not seem to be enhanced by the presence of VP40.

The predominant cytoplasmic localization of VP24 is fairly consistent with a previous study [13] and the lack of a predominant lipid-binding phenotype. We further observed that fluorescent fusion constructs of VP24 can form several punctate structures in some cells although these structures need more careful investigation (Figure 6). In the closely related MARV, a small portion of VP24 was shown to be associated with intracellular membranes via subcellular fractionation and flotation assays; however, the majority of MARV VP24 protein was cytoplasmic [43]. This seems to be in consonance with the similar and earlier observation of EBOV VP24 [12] and the findings in the current study.

While the basis of localization of VP24 to cellular membranes and punctae in the absence of other filovirus proteins is unknown, it is possible that VP24 is integrated into the membrane fraction indirectly such as through a cellular binding partner that is associated with membranes. Indeed, EBOV VP24 was found to interact with sodium-/potassium-transporting ATPase subunit alpha-1 (ATP1A1) in a VP24 cellular interactome study [44], which may explain why VP24 was slightly detected in the plasma membrane (Figure 5) as ATP1A1 is plasma membrane localized. We also cannot completely rule out that VP24 binds a minor cell lipid, as we did not test for this in our assays; however, we included the majority of bioactive lipids involved in similar processes for other viruses and known peripheral protein targets. Further, all lipid-binding studies employed VP24 with a small tag (e.g., HA or His) and, while it is unlikely these small tags preclude lipid binding, this cannot be completely ruled out. It should be noted, however, we attempted VP24 studies without a tag but had challenges in detection with an anti-VP24 antibody. Thus, all studies employed VP24 with a tag for ease of detection, where previous studies employing similar VP24 tagged approaches produced active VP24 [2,22,39,44]. Nonetheless, more details and direct experimental evidence may be needed to draw solid conclusions on the origins of VP24 intracellular targeting. In closing, we provide detailed evidence that VP24 lacks classical lipid-binding properties ascribed to a peripheral or matrix protein suggesting its predominant recruitment into filovirus particles is through NC assembly and protein–protein interactions.

## Figures and Tables

**Figure 1 viruses-12-00869-f001:**
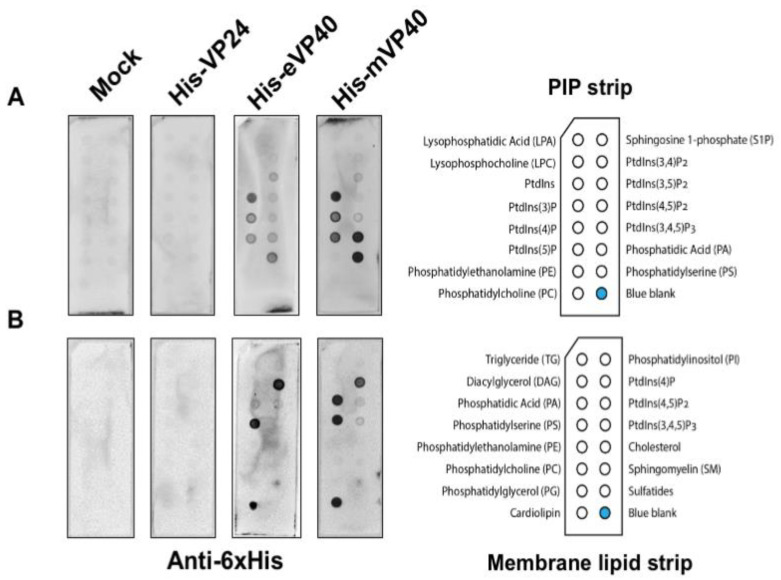
Ebola virus VP24 does not bind to lipids by lipid overlay assay. In this assay, 1 μg/mL of purified recombinant His-VP24, His-eVP40 and His-mVP40 proteins were incubated with PIP lipid strip (**A**) and membrane lipid strip (**B**) overnight at 4 °C. The protein purified from *E. coli* transformed with the empty protein expressing vector was used as the mock control. eVP40 and mVP40 bound to certain anionic lipids as expected while VP24 did not bind to any lipid tested.

**Figure 2 viruses-12-00869-f002:**
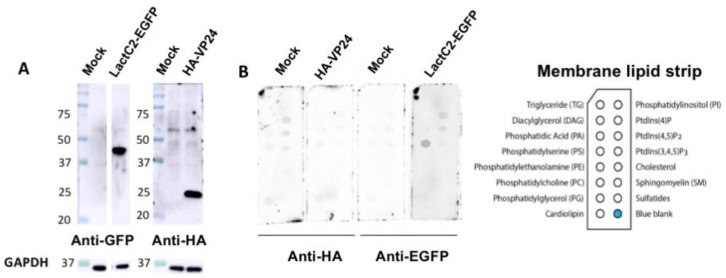
VP24 expressed in HEK293 cells did not bind to lipids. (**A**). The expression of LactC2-EGFP and HA-VP24 were confirmed by Western blot. (**B**). The 80 μg/mL of cell lysate expressing indicated proteins was incubated with membrane lipid strips overnight at 4 °C. The cell lysate of HEK 293 cells with mock transfection was used as a control. HA-VP24 did not bind to any lipids while LactC2-EGFP bound to PS as expected.

**Figure 3 viruses-12-00869-f003:**
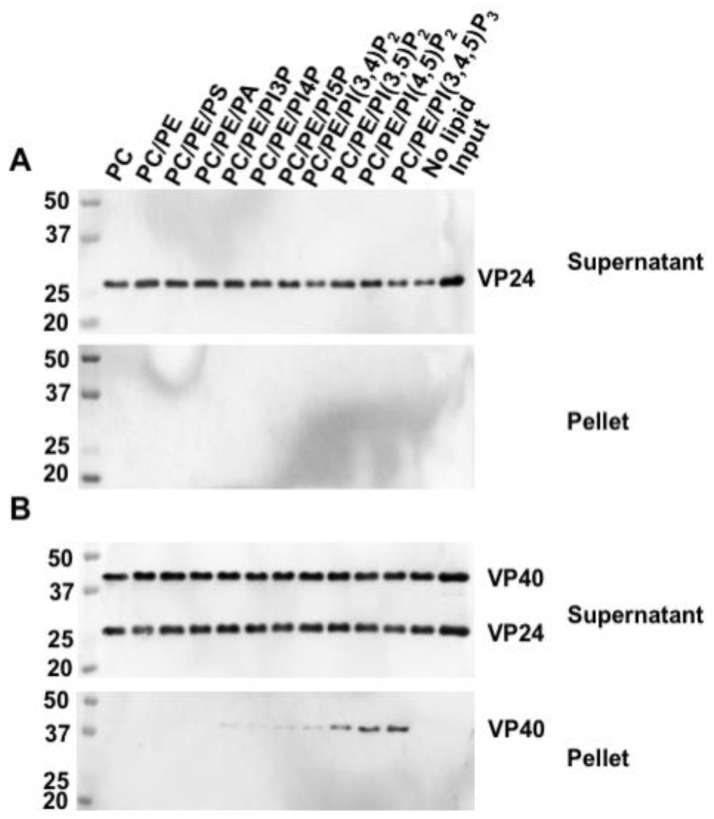
VP24 did not bind to lipids in the presence of eVP40 by protein–liposome sedimentation assay. In this assay, 1 μg of purified VP24 (**A**) or the mixture of 1 μg of purified VP24 and 1 μg of eVP40 (**B**) was (were) incubated with various liposomes for 45 min at room temperature (RT). The protein was detected by anti-His-alkaline phosphatase antibodies in the supernatant and liposome pellet after centrifugation. VP24 alone or along with eVP40 was not associated with any liposome pellet. eVP40 showed up in the expected liposome pellet.

**Figure 4 viruses-12-00869-f004:**
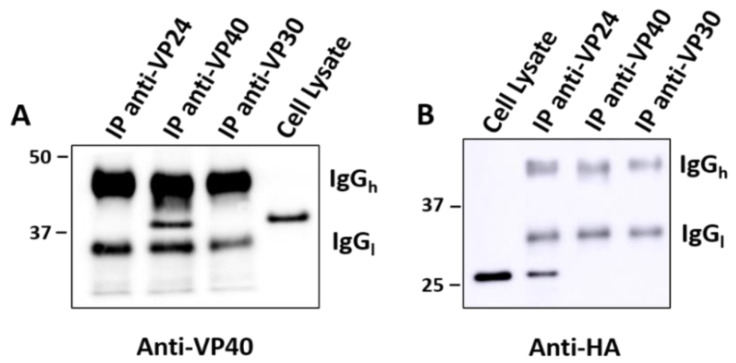
VP24 and VP40 did not interact as evidenced by co-immunoprecipitation (Co-IP). HEK 293 cells were transfected with both HA-VP24 and eVP40 plasmids. Co-IP was performed with anti-VP24, anti-VP40 or anti-VP30 as a control. VP40 and HA-VP24 in IPs were examined by anti-VP40 antibodies (**A**) and anti-HA antibodies (**B**), respectively. No VP24 or VP40 were detected in the IP of anti-VP30 antibodies. VP24 was detected in the IP of anti-VP24 but not in that of anti-VP40. Similarly, VP40 was detected in the IP of anti-VP40 but not in that of anti-VP24. Thus, VP24 and VP40 appear not to interact with each other in cell lysates. IgGh, antibody IgG heavy chain. IgGl, antibody lgG light chain.

**Figure 5 viruses-12-00869-f005:**
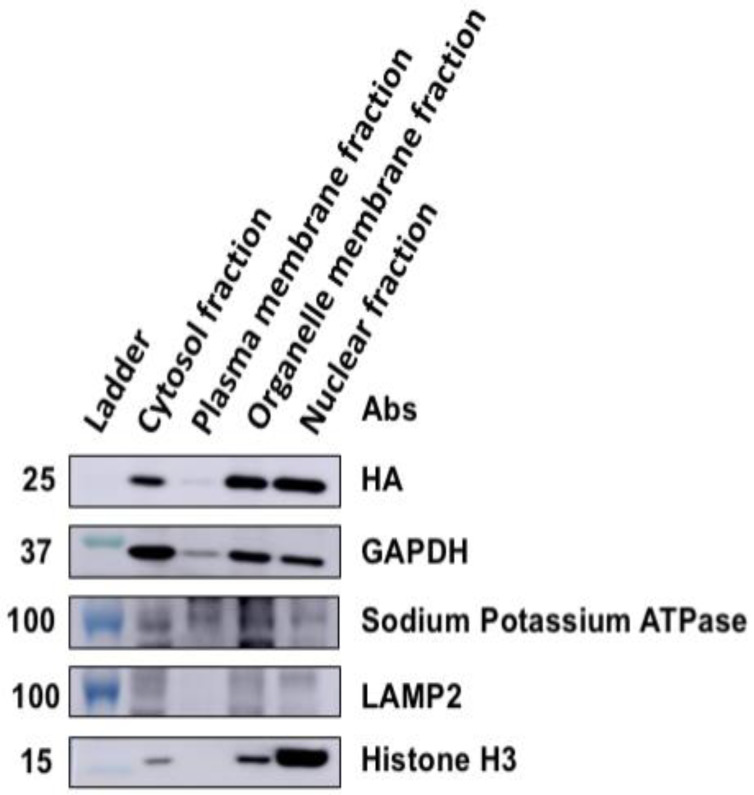
VP24 slightly associated with the plasma membrane fraction from HEK293 cells. The cellular membrane fractionation assay was performed with HA-VP24 detected in the cytosol fraction, organelle membrane fraction and nuclear fraction. Only a trace amount of eVP24 was detected in the plasma membrane fraction.

**Figure 6 viruses-12-00869-f006:**
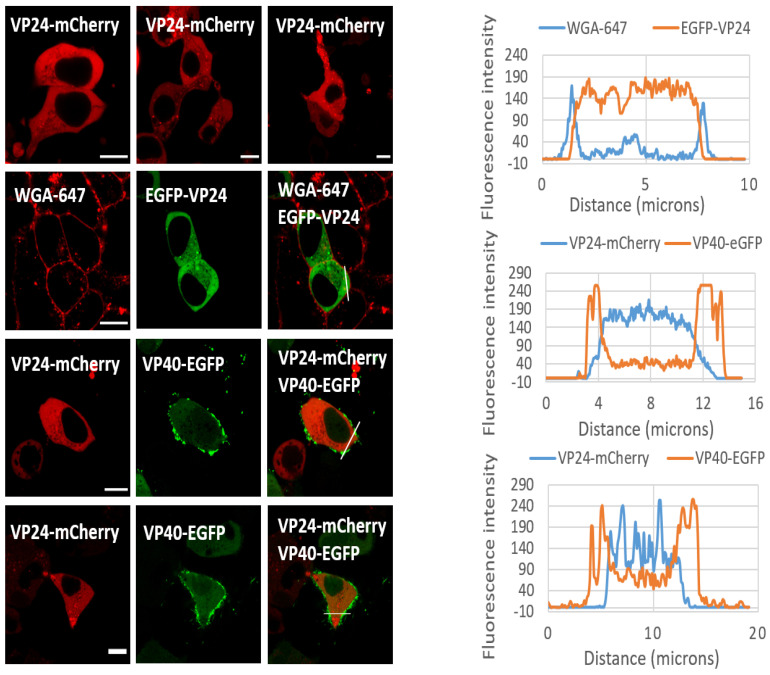
VP24 is not specifically associated with the plasma membrane or VP40. WGA-647 is the plasma membrane marker. EGFP-VP24 or VP24-mCherry were predominantly cytosolic and VP24 did not colocalize with WGA-647 or VP40 specifically. Lines were drawn across the cell to determine the fluorescence intensity of VP24 or VP40 at various points along the line (cytosol or membrane). Scale bar, 10 μm.

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
