# Peer review of "The Minor Matrix Protein VP24 from Ebola Virus Lacks Direct Lipid-Binding Properties"

_viruses, 2020, doi:10.3390/v12080869_

Round 1
Reviewer 1 Report
The manuscript addresses the lipid-binding properties, if any, of the VP24 protein. The experiments have been well planned and appear to have been carried out correctly. My main objection to them arises from the lack of binding experiments using tag-free VP24. Although the different tags used do not seem to affect the lipid binding properties of VP40, its effects on VP24 have not been determined. I consider that it is not possible to generalize to (WT-)VP24 the conclusions given in the Discussion and the Title without this type of “control” experiments.
Author Response
First, we thank the reviewer for his or her time and effort in reviewing our paper. We do agree with the reviewer that using untagged proteins in many assays can be optimal so as to replicate the native condition as carefully as possible. However, small tags such as -His, -HA, or -Flag can also be advantageous in order to robustly detect protein in different types of cellular or in vitro assays. Indeed, this type of approach was needed for VP24. We attempted using the commercially available anti-VP24 antibody and struggled mightily with VP24 detection. Further, for some reason still unknown to us, the anti-VP24 antibody demonstrated background binding to lipids in the overlay assay whereas the anti-His antibody did not. Thus, this was the main reason for employing VP24 with small tags in the various assays.
It should also be noted, for importance and relevance in our work as presented here, VP24 with various tags have been shown to retain biological functions. For instance, in reference 39 (Vidal et al. 2019 J. Virol.), HA-VP24 was functional in being SUMOylated. Further, FLAG-VP24 was shown in references 2 (Fanunza et al. 2018 Viruses) and 22 (Reid et al. 2007 J. Virol.) in VP24 assays of inhibition of IFN signaling and inhibiting the NPI-1 interactions with STAT1, respectively. GFP-VP24 (a larger tag) has also been used in reference 44 (Garcia-Dorval et al. 2014 J. Proteome Research) to unveil protein-protein interactions of VP24 with the host cell. Thus, the biological functions of VP24 have been intact in other studies employing the small tag (or even the GFP approach).
It should also be noted that the majority of lipid-binding protein studies we know of that have been published use some form of small tag detection in lipid-protein interaction analysis of both human and viral proteins. For instance, see (Stahelin et al. 2007. J. Biol. Chem., Ward et al. 2013 J. Lipid Res., Scott et al. 2020 Biomolecules.). Thus, we've not made changes in the tagged approach at this time due to lack of feasibility in careful detection of untagged VP24 with a reliable antibody.
Reviewer 2 Report
The manuscript by Su and Stahelin examines whether the Ebola virus VP24 proteins, which has been thought to be a minor matrix protein, binds to membranes or to other viral membrane-associated proteins (VP40). Using several independent assays, the authors were unable to detect VP24 binding to a set of bioactive lipids or to the Ebola virus VP40, nor they observed the presence of VP24 in the membrane fraction following subcellular fractionation. In addition, confocal microscopy revealed no colocalization of VP24 with plasma membrane markers. This work thus provides considerable evidence against the notion that VP24 interacts with viral membrane and is a minor matrix protein of Ebola virus.
I have a few concerns related to experimental protocols and quality of data.
- I wonder if extensive washes of lipid strips with a detergent (Tween 20) can remove the spotted lipids. Related to this point, please describe how cellular lysates used in Figure 2 were made. If these were detergent lysates, the above concern regarding removal of lipids from strips remains. If these were just cytosolic extracts, no membrane-associated proteins would be present in it.
- The LactC2 binding to PS (Fig. 2B) is weak. Can a higher concentration of lysate or a recombinantly produced protein be tried?
- Subcellular fractionation results (Fig. 5) are not clean. First, why should VP24 associate with organelle membranes? Second, to my knowledge, GAPDH should not associate with the nucleus. Third, sodium-potassium ATPase and LAMP1 signals are week and appear to be present in at least three out of 4 fractions. This raises concerns regarding the quality of fractionation.
- Why do use DTT in protein-liposome co-sedimentation experiments? Can DTT affect the VP24 structure?
- Please specify ImageJ plugin (if any) used for image analysis.
- Please correct multiple typos and loose language. For instance, “3.2. Mammalian cells expressing VP24 protein does not bind to any lipid either” does not seem appropriate for a subtitle. “Figure 4. VP24 did not interact with VP40 by co-immunoprecipitation” – there is no interaction by Co-IP.
Author Response
We would like to thank the reviewer for taking the time to critically read our manuscript and provide the important feedback. Below we respond to the 6 points raised in the review.
- I wonder if extensive washes of lipid strips with a detergent (Tween 20) can remove the spotted lipids. Related to this point, please describe how cellular lysates used in Figure 2 were made. If these were detergent lysates, the above concern regarding removal of lipids from strips remains. If these were just cytosolic extracts, no membrane-associated proteins would be present in it.
Response: This is a fair point raise by the reviewer but we should point out that this is a standard protocol for lipid overlay assays (Shirey et al. 2016 Anal. Biochem. see also Avanti Polar Lipids and Echelon Biosciences who sell such assay kits). Tween-20 is commonly used in the lipid overlay assay to reduce nonspecific binding. It doesn't remove the spotted lipids as hundreds of not thousands of papers have used such approaches with reproducible binding shown for many lipid binding domains and proteins.
For the cell lysate prepared in the Figure 2, cells were lysed in the NP-40 lysis buffer (50 mM Tris-HCl, pH 7.4, containing 150 mM NaCl, 0.1% NP-40, 1x Halt protease inhibitors). NP-40 does not affect lipid binding as suggested in the positive control.
- The LactC2 binding to PS (Fig. 2B) is weak. Can a higher concentration of lysate or a recombinantly produced protein be tried?
Response: This is also a great question from the reviewer as few have measured binding of protein expressed in mammalian cells using lipid binding or overlay assays. The relatively weak binding by Lact C2 is expected here as the PS on the lipid strip is 16:0 PS which was demonstrated previously to have lower binding affinity to LactC2 compared with unsaturated PS species (Del Vecchio and Stahelin, 2018, J Bioenerg Biomembr. 50(1): 1–10).
- Subcellular fractionation results (Fig. 5) are not clean. First, why should VP24 associate with organelle membranes? Second, to my knowledge, GAPDH should not associate with the nucleus. Third, sodium-potassium ATPase and LAMP1 signals are week and appear to be present in at least three out of 4 fractions. This raises concerns regarding the quality of fractionation.
Response: The reviewer is correct that the fractionation is not very clean but the main goal here was to determine VP24 in the plasma membrane fraction due to the reports and mechanisms previously presenting VP24 as the minor matrix protein. The commercial kit we used is a plasma membrane protein extraction kit, which does separate different cellular fractions. We got the highest quality data we could out of it. The nuclear fraction may be contaminated slightly with the cytosol, but it does not affect the fact we wanted to point out that VP24 was not predominately plasma membrane associated. The organelle membrane association with VP24 was likely as it was shown to be present in the integrant membrane fraction previously [see reference 12]. This may occur through the binding with cellular proteins that associate with organelle membranes. However, the molecular details need to be further investigated.
The ATPase band in membrane fraction is slightly higher than unspecific bands in other fractions. Thus, it is only present in the plasma membrane fraction. For Lamp2, there may indeed be contamination from other fractions. If the reviewer still finds the data an issue, we can consider just presenting the cytosol and plasma membrane fractions as the goal of the manuscript revolves around VP24 and lipid-interactions relevant to matrix assembly at the plasma membrane.
4. Why do use DTT in protein-liposome co-sedimentation experiments? Can DTT affect the VP24 structure?
Response: This is a fair point. First, DTT is often used in protein purification and storage as a reducing agent (to prevent protein oxidation as well as any cys-cys bond formation across free cysteine residues in the proteins). DTT is also commonly used in lipid related assay to prevent lipid oxidation. Thus, we and other have routinely employed DTT in these types of lipid binding assays. For additional details, see Magdalena et al. Methods Mol. Biol. 1009:261-71. Further, the crystal structure of the Sudan virus VP24 was solved in the presence of 1 mM DTT (see reference 23) so we don't expect DTT to have major issues on structure of lipid binding of VP24.
- Please specify ImageJ plugin (if any) used for image analysis.
Response: We did not use any ImageJ plugin. A line was drawn on the images and then the plot profile function was used to measure the fluorescence signal along the line. This has been clarified in the text.
- Please correct multiple typos and loose language. For instance, “3.2. Mammalian cells expressing VP24 protein does not bind to any lipid either” does not seem appropriate for a subtitle. “Figure 4. VP24 did not interact with VP40 by co-immunoprecipitation” – there is no interaction by Co-IP.
Response: Thanks for pointing this out. We have revised these points and other issues we found in revising the manuscript. The revised version has used track changes.
Round 2
Reviewer 1 Report
The title information is too much cathegoric and not supported by the designed experiments where, for example, the influence of tags in the lipid binding assays has not been proven. I recommend the authors to modify it by a more realistic one.